# Living with a left ventricular assist device: Capturing recipients experiences using group concept mapping software

Anita L. Slade[1,2,3,4]*, Margaret E. O'Hara[5,6], David Quinn[5], Laura Marley[5], Stephen Griffith[1,7], Melanie Calvert[1,2,3,4,8], M. Sayeed Haque[9], Hoong Sern Lim[5], Philippa Doherty[5], Deirdre A. Lane[10,11]

1 Centre for Patient Reported Outcomes Research, Institute of Applied Health Research, University of Birmingham, Edgbaston, Birmingham, United Kingdom, 2 National Institute for Health Research (NIHR) Applied Research Centre West Midlands, Birmingham, United Kingdom, 3 National Institute for Health Research Birmingham Biomedical Research Centre, University of Birmingham, Birmingham, United Kingdom, 4 National Institute for Health Research Surgical Reconstruction and Microbiology Research Centre, University of Birmingham, Birmingham, United Kingdom, 5 Queen Elizabeth Hospital, University Hospital Birmingham NHS Foundation Trust, Birmingham, United Kingdom, 6 Institute of Translational Medicine, Queen Elizabeth Hospital, University Hospital Birmingham NHS Foundation Trust, Birmingham, United Kingdom, 7 Patient Research Partner, Carmarthenshire, United Kingdom, 8 Birmingham Health Partners Centre for Regulatory Science and Innovation, University of Birmingham, Birmingham, United Kingdom, 9 Institute of Applied Health Research, University of Birmingham, Birmingham, United Kingdom, 10 Department of Cardiovascular and Metabolic Medicine and Liverpool Centre for Cardiovascular Science, University of Liverpool and Liverpool Heart and Chest Hospital, Liverpool, United Kingdom, 11 Department of Clinical Medicine, Aalborg University, Aalborg, Denmark

* a.l.slade@bham.ac.uk

**Data Availability Statement:** All relevant data are within the article and its Supporting Information files.

## Abstract

### Background

Left ventricular assist device (LVAD) implantation significantly impacts on a recipient's symptoms and quality of life. Capturing their experiences and post implant journey is an important part of clinical practice, research and device design evolution. Patient reported outcome measures (PROMs) are a useful tool for capturing that experience. However, patient reported outcome measures need to reflect recipients' experiences. Discussions with a patient partner group found that none of the frequently used cardiology PROMs captured their unique experiences.

### Aims

To capture the experiences and important issues for LVAD recipients. Develop a conceptual map of domains and items that should be reflected in patient reported outcomes.

### Methods

Group concept mapping (GCM) web-based software was used to remotely capture and structure recipients' experiences across a wide geographical area. GCM is a semi-quantitative mixed method consisting of 3 stages: item generation, item sorting and rating

**Funding:** This work was supported by funding from the British Heart Foundation (grant number PG/18/58/33944) awarded to ALS. This work was also supported by the NIHR, awarded to MC. (This paper presents independent research supported by the National Institute for Health and Care Research Birmingham Biomedical Research Centre at the University Hospitals Birmingham National Health Service Foundation Trust and the University of Birmingham).

**Competing interests:** The authors have declared no competing interests exist.

**Abbreviations:** AHF, Advanced Heart Failure; FDA, U.S. Food and Drug Administration; GCM, Group Concept Mapping; HRQoL, Health Related Quality of Life; LVAD, Left Ventricular Assist Device; MDS, Multi-Dimensional Scaling; PROM, Patient Reported Outcome Measure; PPI, Patient and Public Involvement.

(importance, relevance and frequency). Patient partners were involved in all aspects of the study design and development.

## Results

18 LVAD recipients consented to take part. 101 statements were generated and multi-dimensional scaling, and hierarchical cluster analysis identified 9 clusters. Cluster themes included: Activities, Partner/family support, Travel, Mental wellbeing, Equipment and clothing, Physical and cognitive limitations, LVAD Restrictions, LVAD Challenges and positive impact of the LVAD (LVAD Positives). LVAD Positives were scored highest across all the rating variables, e.g., frequency (2.85), relevance (2.44) and importance (2.21). Other domains rated high for importance included physical and cognitive limitations (2.19), LVAD restrictions (2.11), Partner/family support (2.02), and Equipment and clothing (2.01).

## Conclusion

Online GCM software facilitated the inclusion of geographically dispersed recipients and provided useful insights into the experiences of LVAD recipients. The conceptual framework identifies important domains and items that should be prioritised and included in patient reported outcomes in future research, LVAD design evolution, and clinical practice.

## Background

Globally, the number of people living with heart failure is increasing, currently affecting approximately 26 million people [1]. Advanced heart failure is associated with high mortality rates (50% at 1 year), and a disease burden similar to other chronic diseases [2, 3]. Heart transplantation offers good long term outcomes, but donor shortages limit this therapy to a minority of potential recipients [4]. A left ventricular assist device (LVAD) is one therapeutic option, offered to patients who are eligible for heart transplantation in the UK [5]. However, in the absence of LVAD-related complications, LVAD recipients then join the low priority routine transplant waiting list. As a result, most uncomplicated LVAD recipients live with LVAD therapy for years, akin to so-called "destination therapy" [6].

While many recipients experience an improvement in their symptoms and health related quality of life (HRQoL), LVAD implantation is associated with mortality risks and serious adverse events such as stroke, right ventricular failure, device related infections, and bleeding [4, 7]. Receiving an LVAD also requires substantial lifestyle changes and adaptation, and can cause psychological and social difficulties for the recipient and their families [8, 9]. Managing these issues requires ongoing clinical intervention and support. Anticipation of adverse events and better understanding of the patient journey after implantation may help prevent some of the issues that arise post implant. One study comparing the impact of two LVAD designs on survival and incidence of stroke at two years found significant differences between devices [7]. Therefore, device design evolutions also need to address recipients' and clinicians concerns by demonstrating that they improve HRQoL as well as reducing adverse events or complications for recipients.

Monitoring potential HRQoL changes requires measurement instruments to be sensitive and relevant. Patient reported outcome measures (PROMs) are one way of capturing a recipient's journey, and the impact of the LVAD on their HRQoL [10]. PROMs are self-reported

questionnaires that evaluate a person's health and HRQoL from their perspective. Research has shown that utilizing PROMs as part of a clinical consultation can improve the dialogue between clinicians and patients, enabling timely clinical interventions, and better utilisation of clinical resources [11–13]. Using PROMs can also improve patient outcomes by identifying potential adverse events, reducing side effects, and improving survival rates [12, 14, 15].

For PROMs to be effective measurement tools they need to be sensitive and reflective of the issues important to the clinical population. Studies have demonstrated that PROMs developed with input from people with lived experience of the health condition are more sensitive and have superior content validity [16–18]. It is important that PROMs used with LVAD recipients should be underpinned by a conceptual framework supporting content validity and reflecting their important key domains [19, 20]. This approach is supported by the European Medical Agency and the U.S. Food and Drug Administration (FDA). Current FDA guidelines require evidence of this approach when using PROMs to support labelling claims for therapeutics and medical devices [18, 20, 21].

However, a review of PROMs currently used in chronic heart failure and mapped against FDA guideline criteria found they were unsuitable for supporting labelling claims [22]. A systematic review also found few studies where PROMS were developed with LVAD recipient input or addressing their psychological issues [23].

These deficiencies were supported by our discussions with LVAD recipients as part of a public and patient involvement (PPI) group [24]. The group suggested that many of the symptoms captured by cardiac-specific measures were no longer relevant to them. Other generic PROMs did not address some of the unique and problematic aspects of their lives, such as dealing with the LVAD equipment and restrictions, psychological impacts, and side effects. The consensus was that these unique issues need to be addressed by a PROM which reflected their experiences. Therefore, the objectives of this current study were: (1) identify the lived experience of LVAD recipients and the impact of living with an LVAD on HRQoL; (2) identify the range of issues, domains and items that were most important to LVAD recipients and (3) improve research and clinical practice by ensuring LVAD recipient voices were reflected in any PROMS used in clinical consultations, research, and LVAD design evolution.

## Methods

### Recruitment

Participants were recruited from one of the UK's national heart transplant centres covering a wide cohort of ethnic groups, and a large geographical area incorporating urban and rural areas. Consent to contact was obtained from the LVAD co-ordinators during routine clinical appointments. Electronic or hard copies of the patient information sheets, demographic questionnaire and consent forms were then sent to the recipient by the research team (based on patient preference). Written or electronic consent was obtained depending on participants preferences for contact. Ethical approval was given by the NHS Health Research Authority (Ref No. 19/WM/0120). Inclusion criteria were all current or previous LVAD recipients aged ≥18 years with access to the internet. Patients with insufficient English, severe communication problems, terminally ill or with severe cognitive or mental health problems were excluded. Participants unable to access the internet were offered the chance to participate in future qualitative interviews. During routine clinical appointments, patients who met the inclusion criteria were approached by LVAD coordinators and if interested in the project, signed a consent-to-contact form. This was sent to the research team, and depending on their preference, electronic or hard copies of the participant information sheet, consent form, and demographic questionnaire were sent to patients.

## Group concept mapping

Group concept mapping (GCM) is a mixed methods approach which can capture and structure recipients' experiences of living with an LVAD. Cluster maps and the rating of generated statements facilitates the identification of key concepts, important domains and statements within each cluster [25, 26]. This structured method of mapping statements from the GCM exercises onto a conceptual framework can be used to identify suitable PROMs for use in clinical practice or research. It can also facilitate PROM development or refinement [27–29]. Future qualitative interviews will be used to explore identified domains in depth, and to facilitate further development and refinement of the conceptual framework [19].

The GCM approach recommended by Kane and Trochim (2007) was used. In the UK LVAD implantation is done in national heart transplant centres which cover a very large geographical area. Data were collected and analysed using the Groupwisdom™ online platform and concept mapping software [25, 26, 30]. Using an online platform allowed us to reach a wider range of LVAD recipients dispersed over this large geographical location. This was especially important as LVAD recipients who have had the implant for any length of time, and without complications infrequently attend clinic appointments, and therefore were less likely to engage with face-to-face GCM discussions. This was also important in order to accommodate COVID restrictions. Participants who consented to take part were given a unique ID, password and link to the Groupwisdom™ concept mapping website [30]. The GCM has three stages and participants were sent a link to activities at each stage. Each activity was completed before moving onto the next stage. Participants were sent a reminder if they had not accessed the activity within two weeks of being sent the link.

## Stage 1. Statement generation

Participants were asked to respond to the prompt *'Life with an LVAD means. . .. . ..'.*

Participants generated statements which reflected their experiences in response to the prompt. Once completed, statements from all participants were aggregated and used in the next stage. If more than one idea was identified within a statement, it was split into separate statements. Duplicate ideas were combined or deleted, and statements were checked for literacy and spelling while trying to maintain the participant's voice.

## Stage 2. Sorting

Participants reviewed the aggregated list and individually sorted statements into thematic groups based on their experience. They labelled the groups based on their perceptions of the thematic content of the group and its meaning to them. This sorting exercise underpins the structure of the cluster analysis.

## Stage 3. Rating

In the final stage, participants rated each statement for relevance to them, frequency, and how important they thought the statement was.

Rating questions and scores included: **Relevance**–"Read each statement and tell us how much you feel this statement reflects your own experience?" 1 = Not at all; 2 = Sometimes my experience; 3 = Definitely my experience. **Frequency**–"How often do you feel like this statement?" 1 = Never; 2 = Sometimes; 3 = Frequently; 4 = All of the time. **Importance**–"How important is this statement to you?" 1 = Not important; 2 = Important; 3 = Very important.

The GCM software allows you to identify the comparative importance, relevance and frequency of each cluster, as well as the statements within the cluster. Participants' scores are

averaged for individual statements and then these are averaged across the cluster. This provided a method for prioritising and structuring items, domains and general concepts based on participants preferences [31].

## Data analysis

Groupwisdom[TM] concept mapping software was used to analyse the relationships and commonalities between participants' responses to the sorting activity [30]. Non-metric multidimensional scaling (MDS) and hierarchical cluster analysis were applied to the data to produce visual representations as point and cluster maps. Full methodological details can be found elsewhere [25–27].

Data analysis consisted of two stages: First, MDS with a two-dimensional solution estimated the relationships (distant and proximal) generated from the summed similarity matrix created from participants' choices in stage two [25, 30]. Statements frequently grouped together are located proximally, while less frequently grouped statements are distal to each other. The stress index is a goodness-of-fit statistic and a lower value indicates that the two dimensional x-y configuration is not random, and there is agreement between the final representation and the original similarity matrix [25, 27]. Secondly, utilizing the MDS x-y configuration, an agglomerative hierarchical cluster analysis uses Ward's minimum variance method to optimise cluster merging [25, 30]. Cluster maps produce graphical representations of the similarities and interrelationships between the domains and statements [27]. There can be as many clusters as statements; therefore, the agglomerative method successively merges statements and clusters. The GCM software allows each merged cluster solution to be reviewed after each iteration. A decision was made on whether the amalgamated content was appropriate, and the range of clusters optimised identifiable distinct groups with relevant thematic content. Each merger was labelled using agree (with the merger), tentative and disagree. The point at which disagree was the most likely option was used as a stopping point. This approach is recommended within the concept mapping literature [25–27].

## Public and patient involvement

A patient and public advisory group with experience of living with an LVAD were consulted throughout the study. The group worked with AS to co-create the participant information documents, GCM prompt and wording of the rating questions, as well as iteratively assisting with the design of the GCM platform. They also advised on networking, recruitment and dissemination strategies. SG was an integral part of this group and acted as a patient research partner throughout the study.

## Results

### Participants

Eighteen LVAD recipients consented to take part in the GCM exercise. Four participants did not complete the GCM (reasons for non-completion included: health issues that prevented participation (n = 1), difficulty accessing the online format (n = 1) and reasons unknown (n = 2). The participant who struggled with the online format took part in subsequent qualitative interviews. One participant died (but had completed all but one rating question), as data were anonymised, their data were included in the results. The number of participants who completed the GCM was within recommended limits (n = 10–40) [26]. The median age of participants was 63 (range 40 to 72) years and the median length of time living with an LVAD was 25 months (range 1 to 86 months).

## Stage 1. Statement generation

A total of 101 statements were generated after reviewing for literacy, conceptual redundancy and duplication.

## Stage 2. Sorting

**Points map.**   MDS analysis produced an overall stress index value of 0.32 after 16 iterations. This falls within the expected reference values (0.21 and 0.39) [26, 32]. Points generated from the x-y plots are shown on the points map (Fig 1A) [26].

**Hierarchical cluster solution.**   A nine-cluster solution offered the best range of identifiable and interpretable distinct groups after reviewing a range of clusters (16 to 4 clusters). The cluster concept map represents the overlying hierarchical cluster analysis results onto the points map (Fig 1B). Cluster labels were generated by reviewing the thematic content of each cluster and labels used by participants.

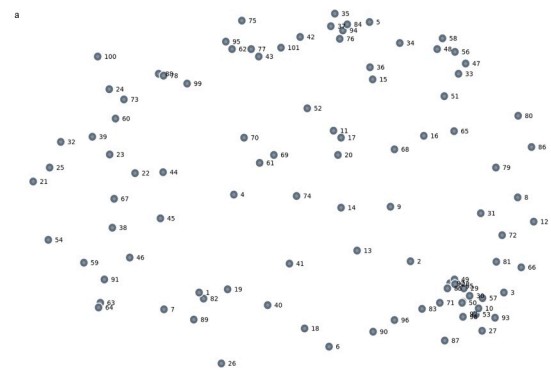

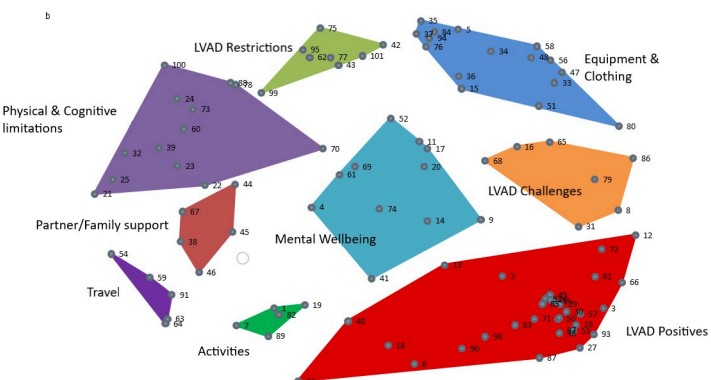

**Fig 1. Points map with overlay of nine cluster solution.** a. Points maps are visual representations of the two-dimensional solution generated from the similarity matrix created from participants choices in stage 2. Frequently grouped statements are located proximally while less frequently grouped statements are distal to each other. b. Cluster maps are visual representations of the hierarchical cluster analysis solution, and represent similarities and inter-relationships between the domains and items. Large clusters represent broader concepts, whereas smaller clusters represent more narrowly focused homogenous domains.

## Cluster descriptions

**Activities.**   Activities consisted of five statements referring to types of activities that participants were able to do since receiving the LVAD. Statements relating to mobility had the highest scores for importance (2.00).

**Partner and family support.**   Five statements related to participants' feelings about their dependency on family/carers for practical and emotional support, and appreciation for that support. The highest rated statements related to gratitude for family support (2.44) and the need to rely on partner/carer for help with dressings (2.13).

**Travel.**   Five statements related to travel, the need to be organised (2.25), and anxieties around security and available care. The need to be organised with the equipment while travelling (2.25) was also scored high for importance.

**Mental wellbeing.**   Eleven statements were included in this cluster relating to issues around the impact of life changing surgery on body image and lifestyle, and difficulties adjusting to the LVAD. Highest rated statements for importance included the impact on mental health of having to come to terms with LVAD restrictions on everyday life (2.22), awareness of mortality (2.11), and challenges with mental health (2.00).

**LVAD challenges.**   Seven statements were included in this cluster and they related to the adjustments required while learning to live with the LVAD, and other peoples' reactions to the equipment. Over 50% (n = 5) of these statements were rated high for relevance, three were rated high for importance including trying to be positive while adjusting to the LVAD (2.33), being more mindful of your body (2.33), and the challenges of adjusting to a new way of life (2.22).

**Equipment and clothing.**   This was the second largest cluster (16 statements) relating to equipment and batteries, dressings and drivelines, and trying to find suitable clothing to wear with the equipment. Nearly 60% of the statements (n = 10) were rated two or higher for importance. Some of the highest rated statements related to finding clothing solutions (2.22), issues with the bags, batteries and controller, the need to be organised with the batteries (2.33), the discipline required to deal with dressings (2.33), and difficulties associated with showering and bathing (2.33).

**Physical and cognitive limitations.**   This was the third largest cluster with 13 statements relating to residual difficulties such as cognitive and memory problems, fatigue, and mobility problems. Nearly 70% (n = 9) of the statements were rated two or higher for importance including some of the highest scores overall relating to fatigue (2.67), and mobility issues (2.50). Other statements were related to not being able to work because of mobility (2.44), limited physical ability (2.44) and mental capacity (2.22).

**LVAD restrictions.**   Eight statements relating to the restrictions imposed on their activities (2.22) and associated with the equipment (2.33). References to lack of spontaneity because of the need to be organised with the equipment (2.33), impact on everyday activities such as shopping (2.22) and worries about getting the equipment wet (2.33) were some of the highest rated statements for importance.

**LVAD positives.**   LVAD Positives had 31 statements associated with the life-saving aspects and positive impacts from the LVAD, and feelings of gratitude in relation to receiving the LVAD. Over 80% (n = 27) of the statements were rated high or very high for importance. These included references to the life-saving aspects of the LVAD (2.67), gratitude to the clinical staff (2.67) and being able to have the LVAD, improvements in health and its impact on HRQoL (2.44) and ability to engage in family activities (2.38).

Bridging values ranging from 0 to 1 (Fig 2 and Table 1) and indicate the extent to which sorting responses by participants were similar in stage 2. Higher values suggest a statement is

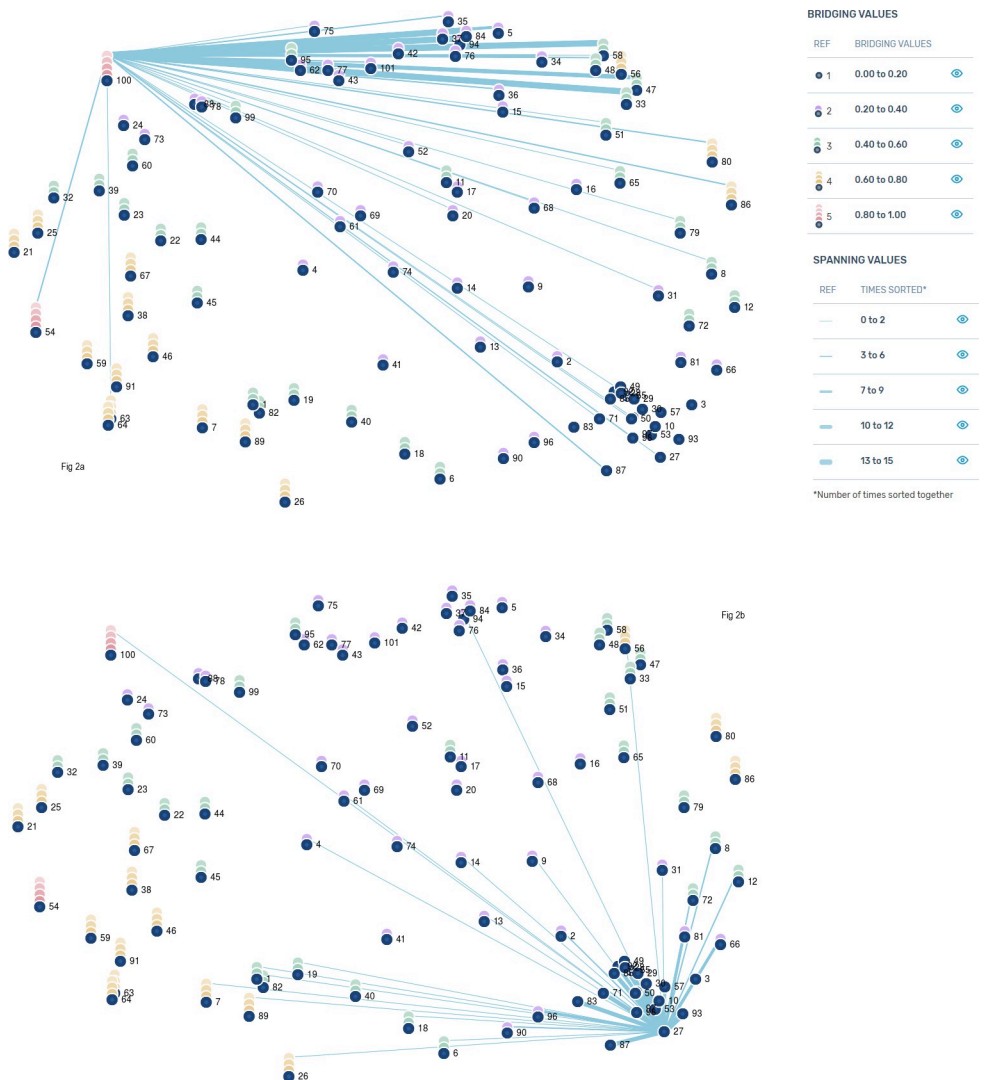

**Fig 2. Bridging values for statements.**

more heterogeneous and has links to broader concepts (Fig 2A). While lower values closer to 0 suggest statements were more homogeneous and more frequently sorted together. These statements can be "anchors" for other statements close to them within the cluster (Fig 2B) [26]. Spanning Values (Fig 2) represent how frequently statements were sorted together, a thicker line demonstrates that other statements were frequently sorted with the selected statement (100 'I have to look after my equipment as my life relies on it' and 27 'Without my LVAD I wouldn't be here').

Some examples of descriptions, bridging values and content of the clusters can be seen in Table 1.

Overall rating scores for the clusters and examples of individual statements within each cluster.

## Stage 3. Rating of individual statements

Rating of the individual statements can be used to identify the relative importance of clusters and individual statements for LVAD recipients. Average scores for clusters and statements

**Table 1. Cluster descriptions and examples of included statements.**

| | Clusters with examples of statements | Mean Scores | | | |
|---|---|---|---|---|---|
| Point ID | Cluster (Total no of statements) | Bridging Values (0–1) | Rating Importance (1 to 3) | Rating Frequency (1 to 4) | Rating Relevance (1 to 3) |
| **1) Activities (5)** | | **0.56 (0.04)*** | **1.37 (0.35)*** | **1.48 (0.49)*** | **1.58 (0.36)*** |
| 1 | I can still play golf using my buggy | 0.53 | 1.00 | 1.00 | 1.18 |
| 82 | I can now walk without any problem | 0.54 | 2.00 | 2.40 | 2.09 |
| **2) Partner/Family support (5)** | | **0.57 (0.09)*** | **2.02 (0.39)*** | **2.36 (0.62)*** | **2.06 (0.41)*** |
| 38 | My partner/carer does my dressings and gives me loads of support. | 0.62 | 2.13 | 2.60 | 2.27 |
| 45 | I no longer have my personal freedom because I need someone with me | 0.52 | 1.50 | 1.70 | 1.67 |
| **3) Travel (5)** | | **0.75 (0.12)*** | **1.88 (0.28)*** | **2.25 (0.47)*** | **2.26 (0.22)*** |
| 54 | It has made me more organized when travelling as the equipment is very important for life now | 1.00 | 2.25 | 2.67 | 2.33 |
| 64 | I am glad I can continue to travel despite the lack of understanding as to what an LVAD is by security staff! | 0.68 | 1.89 | 2.10 | 2.09 |
| **4) Mental Wellbeing (11)** | | **0.32 (0.05)*** | **1.79 (0.33)*** | **2.16 (0.45)*** | **1.92 (0.33)*** |
| 17 | I have had challenges with my mental health | 0.28 | 2.00 | 2.33 | 1.83 |
| 69 | I was extremely grateful at first, but as time goes by, I feel like I am existing and not living anymore | 0.29 | 1.63 | 1.80 | 1.58 |
| **5) LVAD challenges (7)** | | **0.46 (0.12)*** | **1.97 (0.30)*** | **2.47 (0.60)*** | **2.27 (0.37)*** |
| 8 | I am adjusting to a new way of living | 0.51 | 2.22 | 2.67 | 2.67 |
| 16 | Other people's reaction to the kit can sometimes be upsetting, but I have found ways round that, and my own mental picture of myself. | 0.39 | 1.56 | 1.60 | 1.64 |
| **6) Equipment and clothing (16)** | | **0.38 (0.13)*** | **2.01 (0.28)*** | **2.67 (0.43)*** | **2.41 (0.33)*** |
| 5 | You have to try to find clothing solutions to carry all the equipment. | 0.22 | 2.22 | 2.80 | 2.50 |
| 58 | I am used to the batteries and controller now. | 0.49 | 2.22 | 3.40 | 2.82 |
| **7) Physical/cognitive limitations (13)** | | **0.50 (0.16)*** | **2.19 (0.31)*** | **2.81 (0.45)*** | **2.25 (0.32)*** |
| 24 | I am unable to work because of a lack of energy | 0.38 | 2.33 | 3.10 | 2.58 |
| 70 | I sometimes struggle to remember things | 0.29 | 2.00 | 2.40 | 2.27 |
| **8) LVAD restrictions (8)** | | **0.35 (0.09)*** | **2.11 (0.21)*** | **2.66 (0.35)*** | **2.32 (0.15)*** |
| 42 | I have to think about what I need when I leave home, even for 10 minutes. | 0.29 | 2.33 | 2.90 | 2.27 |
| 77 | There are restrictions to normal activities. | 0.30 | 2.22 | 3.10 | 2.55 |
| **9) LVAD Positives (31)** | | **0.22 (0.18)*** | **2.21 (0.34)*** | **2.84 (0.47)*** | **2.44 (0.39)*** |
| 28 | I am happy with my LVAD. | 0.11 | 2.33 | 3.30 | 2.67 |
| 53 | The LVAD saved my life. | 0.04 | 2.44 | 3.33 | 2.82 |

*Averaged scores across the cluster (SD)

(Table 1), allow comparisons across and within clusters for the different rating scales. Pattern match figures (Fig 3) provide visual comparisons of average cluster scores across the different rating variables.

## Conceptual framework

Using information provided by the GCM enabled us to review the content and scoring of individual clusters and statements, and facilitated the development of a conceptual framework for the important items and domains [19, 27]. This conceptual framework can be used to ensure that selection of patient reported outcomes used in clinical practice or research reflect the concerns of LVAD recipients.(Fig 4) Individual statements can be mapped to items within PROMs to ensure they are capturing the issues that are considered important for recipients.

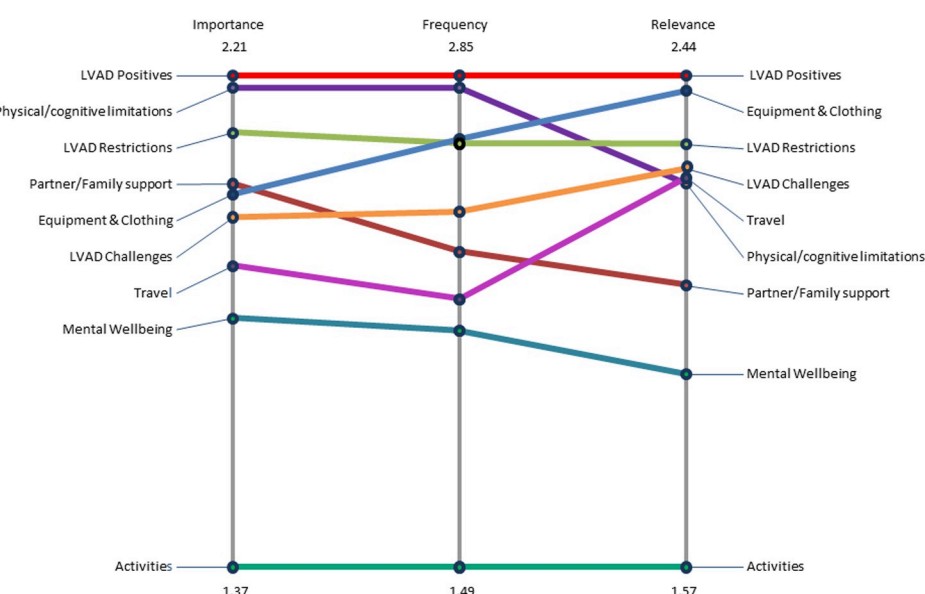

**Fig 3. Average rating scores pattern match by cluster.** Average cluster scores are shown at the top and bottom of the ladder. Clusters are positioned on the vertical axis and vertical lines represent each rating variable. Horizontal lines join the cluster scores within each rating variable for comparison. This shows the relative importance of clusters for recipients.

## Discussion

Using GCM provided a structured approach to the development of a conceptual framework, allowing us to identify the relative importance of the different domains and items for LVAD

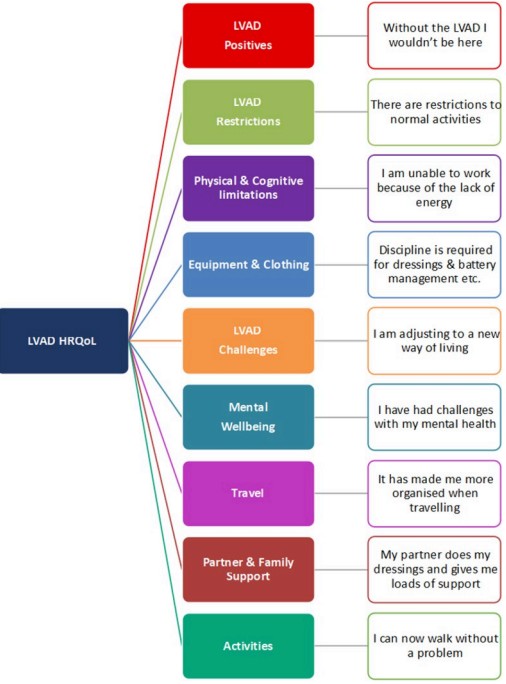

**Fig 4. LVAD HRQoL conceptual framework.** Conceptual framework based on clusters and statements within identifying key domains and items that are important to LVAD recipients.

recipients [19, 25]. A clear conceptual framework should underpin PROM development and evaluation if they are to represent the lived experiences of LVAD recipients [33]. Having a clear concept is important for use in clinical trials, design evolution or research as it places the patient and their experience at the heart of clinical trials, healthcare and research [33]. GCM has several advantages when developing or identifying suitable PROMs for use with this group. It provides a clearly defined method of eliciting the content required for any PROMs. It also allows identification of domains or items that are relatively unimportant to participants and this is useful when trying to minimise irrelevant content [28].

The LVAD positives cluster was rated highest across all the rating values and was viewed as important and relevant; the bridging values suggest this is a homogenous set of items which reflect the need to make this a focus for any PROMs being used with LVAD recipients. Recipients' positive experiences with the LVAD is a valuable insight as positive experiences can sometimes be overlooked by clinicians who may inadvertently concentrate on the negative impacts of an intervention. Identifying LVAD recipients whose experiences do not reflect this positive stance may also indicate that clinical intervention or additional support is required. It might also be useful to monitor whether positive experiences increase or decrease over time as recipients adapt to life with an LVAD.

Some participants were still experiencing physical and cognitive challenges as demonstrated by cluster seven which was scored second highest for importance. This was one of the largest clusters and covered residual issues such as cognitive function, fatigue and limitations in physical capacity. This suggests that while the LVAD does have a positive impact there are still some outstanding issues that might need to be addressed. Not surprisingly, Equipment and Clothing, and LVAD restrictions were also scored high for importance. Given the unique issues that LVAD recipients experience, this reflects the importance of ensuring any PROMs used with this group reflect these experiences, otherwise the full extent of improvements or deterioration in HRQoL and adjustment to life with an LVAD will not be captured.

Individual statements were also highlighted as important within the clusters. Within the LVAD positives cluster, one of the highest scoring statements was "*I feel grateful that it has prolonged my life*". One of the highest scoring statements for equipment was around the need to be disciplined with the equipment "*Discipline is required for dressings, battery management, showering, logbook completing. My life depends on this discipline*". Having access to this detailed data can reveal the relative importance of experiences, and these issues would need to be reflected in any PROMs used. These insights can facilitate the generation of outcomes that matter to recipients and facilitate further in-depth exploration through qualitative interviews.

While it could be argued that the sample completing the exercise was small and the opportunity for the development of the issues in depth is limited, the number completing the GCM is within the recommended range for concept mapping studies, and reflects the small number of LVAD recipients [26].

Using the online software to remotely capture their experiences allowed us to recruit participants from a wide geographical area. In the UK, LVAD implant services are few and dispersed across the country. Consequently, patients often have to travel long distances to attend hospital appointments. Access to this online format is especially important in this post-COVID era when many hospital consultations have moved online. One of the limitations of this approach is that it does require a certain degree of computer and language literacy for participants to take part; this was evident as one participant struggled to access the site even with support from the research team. However, this participant was offered the opportunity to take part in qualitative interviews so their experiences can still inform our understanding of their issues. Limiting the language to English may mean additional cultural perspectives may have been lost. One limitation of the current study is the potential for gender bias as no women agreed to

take part in this part of the research programme. This maybe a reflection of the small proportion of women (<20%) who receive an LVAD [34]. However, female participants agreed to take part in the qualitative interviews and we will explore whether female insights differ substantially from their male counterparts.

One of the strengths of this study is the involvement of the PPI group who have contributed throughout the study providing initial guidance on the need for a specific PROM, to the design and development of the online GCM process, providing unique input and insights at different stages of the project. One of the group (SG) has acted as a co-investigator throughout the project.

The results of the GCM can be used to develop a deeper understanding of the issues for LVAD recipients. The domains and statements were also used to develop a topic guide for qualitative interviews with LVAD recipients. Relying on literature alone to develop a topic guide might have missed the personal and unique nature of LVAD recipients' experiences shared during this GCM process.

## Conclusion

The online GCM exercise has produced a conceptual map of key areas that are important to LVAD recipients. The GCM will support the development of semi-structured qualitative interviews to allow in-depth discussion of the key points raised by participants. Together, this information will support prioritised patient reported outcome measurement in future research, design evolution and clinical practice with LVAD recipients.

## Supporting information

**S1 Fig. Importance point rating map.** Points map showing which statements were considered important by LVAD recipients. *Scoring Importance*: 1 = Not important; 2 = Important; 3 = Very important.
(TIF)

**S2 Fig. Relevance points rating map.** Points map showing which statements most reflected recipients' experiences. *Scoring for relevance*: 1 = Not at all; 2 = Sometimes my experience; 3 = Definitely my experience.
(TIF)

**S3 Fig. Frequency points rating map.** Points map showing which statements recipients thought frequently reflected their experiences. *Scoring for Frequency*: 1 = Never; 2 = Sometimes; 3 = Frequently; 4 = All of the time.
(TIF)

**S1 Data.**
(XLSX)

**S1 Protocol.**
(PDF)

## Acknowledgments

We would like to thank Richard Lamens, Bill Webb and members of the LVAD PPI group for co-creating the participant information documents, GCM prompt and wording of the rating questions, as well as assisting with the design of the GCM platform and advising us on recruitment, networking, and dissemination strategies. We would also like to thank Queen Elizabeth

Hospital Birmingham LVAD Coordinators for help with the recruitment of participants to the study. We would like to thank Anita Walker for administrative support, and Barbara Torlinska for additional statistical advice.

**Disclaimers:** The views expressed in this article are those of the author(s) and not necessarily those of the funders.

## Author Contributions

**Conceptualization:** Anita L. Slade, David Quinn, Laura Marley, Stephen Griffith, Melanie Calvert, M. Sayeed Haque, Hoong Sern Lim, Deirdre A. Lane.

**Data curation:** Anita L. Slade, M. Sayeed Haque.

**Formal analysis:** Anita L. Slade.

**Funding acquisition:** Anita L. Slade, Margaret E. O'Hara, David Quinn, Laura Marley, Stephen Griffith, Melanie Calvert, M. Sayeed Haque, Hoong Sern Lim, Deirdre A. Lane.

**Investigation:** Anita L. Slade, Margaret E. O'Hara, David Quinn, Laura Marley, Stephen Griffith, M. Sayeed Haque, Deirdre A. Lane.

**Methodology:** Anita L. Slade, Margaret E. O'Hara, David Quinn, Laura Marley, Stephen Griffith, Melanie Calvert, M. Sayeed Haque, Hoong Sern Lim, Philippa Doherty, Deirdre A. Lane.

**Project administration:** Anita L. Slade, Philippa Doherty.

**Resources:** Anita L. Slade, Margaret E. O'Hara, Laura Marley, Hoong Sern Lim, Philippa Doherty.

**Software:** Anita L. Slade.

**Supervision:** Anita L. Slade, Melanie Calvert, Deirdre A. Lane.

**Validation:** Anita L. Slade, Deirdre A. Lane.

**Visualization:** Anita L. Slade, Margaret E. O'Hara, Deirdre A. Lane.

**Writing – original draft:** Anita L. Slade.

**Writing – review & editing:** Anita L. Slade, Margaret E. O'Hara, David Quinn, Laura Marley, Stephen Griffith, Melanie Calvert, M. Sayeed Haque, Hoong Sern Lim, Philippa Doherty, Deirdre A. Lane.

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
