## [Decision Letter · Decision Letter 0]

8 Mar 2022

PONE-D-21-38323

Living with a Left Ventricular Assist Device: capturing recipients experiences using group concept mapping software<o:p></o:p>

PLOS ONE

Dear Dr. Slade,

Thank you for submitting your manuscript to PLOS ONE. After careful consideration, we feel that it has merit but does not fully meet PLOS ONE’s publication criteria as it currently stands. Therefore, we invite you to submit a revised version of the manuscript that addresses the points raised during the review process. Two reviewers have read your work. Please consider submitting a revised version of your work based on comments of reviewers.

We look forward to receiving your revised manuscript.

Kind regards,

Salil Deo

Academic Editor

PLOS ONE

Journal Requirements:

“This work was supported by funding from the British Heart Foundation (PG/18/58/33944). This paper presents independent research supported by the National Institute of Health Research Birmingham Biomedical Research Centre at the University Hospitals Birmingham National Health Service Foundation Trust and the University of Birmingham. The study funders did not have any role in the study design; the collection, analysis, and interpretation of the data; the writing of the report; or the decision to submit the article for publication.

ALS was supported by funding from: The British Heart Foundation, National Institute for Health Research Invention for Innovation, Medical Research Council, National Institute for Health Research MedTech and INvitro diagnostics Co-operative–Trauma management, National Institute for Health Research Clinical Research Network; National Institute For Health Research  Birmingham Biomedical Research Centre and National Institute For Health Research Surgical Reconstruction and Microbiology Research Centre at the University of Birmingham and University Hospitals Birmingham National Health Service Foundation Trust outside the submitted work.

MC is a National Institute for Health Research Senior Investigator and receives funding from the National Institute for Health Research Birmingham Biomedical Research Centre, the National Institute for Health Research Surgical Reconstruction and Microbiology Research Centre and National Institute for Health Research Applied Research Collaboration West Midlands at the at the University of Birmingham and University Hospitals Birmingham National Health Service Foundation Trust, Health Data Research United Kingdom, Innovate United Kingdom (part of United Kingdom Research and Innovation), Macmillan Cancer Support, Health Foundation, UCB Pharma and GSK. MC has received personal fees from Astellas, Takeda, Merck, Daiichi Sankyo, Glaukos, GlaxoSmithKline and the Patient-Centred Outcomes Research Institute (PCORI) outside the submitted work.

DL has received investigator-initiated educational grants (paid to the institution) from Bristol-Myers Squibb (BMS) and personal fees from Boehringer Ingeheim, Bayer, BMS/Pfizer, and Daiichi-Sankyo outside the submitted work.”

We note that you have provided additional information within the Funding Section that is not currently declared in your Funding Statement. Please note that funding information should not appear in other areas of your manuscript. We will only publish funding information present in the Funding Statement section of the online submission form.

“This work was supported by funding from the British Heart Foundation (PG/18/58/33944). This paper presents independent research supported by the National Institute of Health Research Birmingham Biomedical Research Centre at the University Hospitals Birmingham National Health Service Foundation Trust and the University of Birmingham.”

4. Please upload a new copy of Figure 1 as the detail is not clear. Please follow the link for more information: " ext-link-type="uri" xlink:type="simple">https://blogs.plos.org/plos/2019/06/looking-good-tips-for-creating-your-plos-figures-graphics/"
https://blogs.plos.org/plos/2019/06/looking-good-tips-for-creating-your-plos-figures-graphics/"

Additional Editor Comments (if provided):

Please consider revising your work based on the reviewer's comments provided.

Reviewers' comments:

Reviewer's Responses to Questions

**Comments to the Author**

1. Is the manuscript technically sound, and do the data support the conclusions?

Reviewer #1: Yes

Reviewer #2: Yes

2. Has the statistical analysis been performed appropriately and rigorously? 

Reviewer #1: Yes

Reviewer #2: Yes

3. Have the authors made all data underlying the findings in their manuscript fully available?

Reviewer #1: Yes

Reviewer #2: Yes

4. Is the manuscript presented in an intelligible fashion and written in standard English?

Reviewer #1: Yes

Reviewer #2: Yes

5. Review Comments to the Author

Reviewer #1: The authors present the results of a quite sophisticated study of a small group (18) of LVAD recipients who are being followed at one of the Heart Failure Canters in UK. The study targets eliciting and evaluating personal experience of living with an LVAD using Concept Group Mapping (GCM) approach. The participants were first asked to generate statements reflecting such experience in several categories. Such statements were then grouped, and the participants have rated those in three categories of relevance, frequency, and importance. All responses were then put through quite complex statistical analysis which evaluated several clusters and allowed comparisons between the clusters and inside each cluster using scores generated for such comparisons. The authors claim that this approach offers important advantage to other Patient Related Outcome Measures( PROMs) often used in evaluating outcomes of LVAD implantation or other life-modifying procedures. The main such advantage comes from the fact that the data are generated by allowing the actual recipients of the treatment ( such as LVAD) to make statements describing different aspects of their life with the LVAD in their own words rather than answer prepared questions. There is data in the literature that the most accurate information regarding life with the LVAD comes from those who live with the LVAD. In addition, GCM approach has been successfully applied for concept generation and evaluation in the past. It does seem to have allowed the authors to benefit from working with the data generated by a small number of participants. Overall, the findings of the study do contribute some additional information as to is important to the LVAD recipients at which stage of their with the LVAD. However, it is hard to imagine that this information by itself would influence the direction of development of the new generations of LVADs.

Reviewer #2: The implantation of a Left Ventricular Assist Device (LVAD) changes the life of both the patient and their family. Despite clinical improvement in most cases, the user is faced with equipment that imposes restrictions and changes in their image as a result of the surgery and the driveline. Beyond the usual knowledge about LVAD functioning and device-associated complications (such as infections, bleeding, thrombosis, or hemolysis), the attending physician should try to address the patient's feelings about their device. This manuscript does an excellent job of putting the user experience with LVADs at the center of the research, making it possible to identify domains and items that can be used in the development of Patient-Reported Outcome Measures (PROMs) in future research. Using the online Group Concept Mapping (GCM) software to identify these domains and items was also a great success for the authors, as it allowed people using LVADs who lived in different geographic regions to collaborate with the research. In addition to allowing the creation of statements, the GCM allows the research participant to sort and rate them, making it possible to identify what should be prioritized in future PROMS.

Despite the good work done by the investigation team, we were able to list some recommendations to improve the understanding of the information generated by the research.

Figures and Tables:

· Figure 1A: Although the figure looks solid, the language of its legend is not clear, making it difficult to understand. I advise you to put some details on the interpretation of the legend as found in Line 209: "Statements frequently grouped together are located proximally, while less frequently grouped 210 statements are distal to each other."

· Figure 1B: This figure represents the clusters well; however, we suggest that each cluster had a different color to facilitate visualization. It’s interesting to add the description about the size of the clusters found between lines 258 - 260.

· Figure 2: This figure is not clear. We suggest detailing a little more about bridging values in the legend (similar between lines 319 and 325). Also, the lines have similar colors, but different thicknesses, so you could be specified what each line thickness is and there are different color points without explanation.

· Figure 3: This figure is the most important, because it shows the importance, frequency, and relevance of each cluster. My suggestion is to put each cluster and line with different colors.

·

6. PLOS authors have the option to publish the peer review history of their article (what does this mean?). If published, this will include your full peer review and any attached files.

Reviewer #1: **Yes: **Yakov Elgudin, MD, PhD

Reviewer #2: No

---

## [Author Response · Author response to Decision Letter 0]

8 Jul 2022

Reference: PONE-D-21-38323

Dear Dr Deo,

Thank you for your positive review of our paper entitled “Living with a Left Ventricular Assist Device: capturing recipient’s experiences using group concept mapping software” and the opportunity to revise the manuscript. We have provided a point-by-point response to the Reviewers’ comments (see ‘Response to Reviewers” document). Amendments have been made in the manuscript and tracked for ease of review. Responses to additional changes requested by the Editorial team are provided below. 

Financial disclosure: This has been removed from the manuscript and is provided below.

This work was funded by the British Heart Foundation (PG/18/58/33944). The study funders did not have any role in the study design, the collection, analysis, or interpretation of the data, the writing of the report, or the decision to submit the article for publication.

ALS was funded for work on this project by the The British Heart Foundation. 

ALS also reports support from National Institute for Health Research Invention for Innovation, Medical Research Council, National Institute for Health Research MedTech and INvitro diagnostics Co-operative–Trauma management, National Institute for Health Research Clinical Research Network; National Institute For Health Research Birmingham Biomedical Research Centre and National Institute For Health Research Surgical Reconstruction and Microbiology Research Centre at the University of Birmingham and University Hospitals Birmingham National Health Service Foundation Trust outside the submitted work.

MC is a National Institute for Health Research Senior Investigator and receives funding from the National Institute for Health Research Birmingham Biomedical Research Centre, the National Institute for Health Research Surgical Reconstruction and Microbiology Research Centre and 

National Institute for Health Research Applied Research Collaboration West Midlands at the at the University of Birmingham and University Hospitals Birmingham National Health Service Foundation Trust, Health Data Research United Kingdom, Innovate United Kingdom (part of United Kingdom Research and Innovation), Macmillan Cancer Support, Health Foundation, UCB Pharma and GSK. MC has received personal fees from Astellas, Takeda, Merck, Daiichi Sankyo, Glaukos, GlaxoSmithKline and the Patient-Centred Outcomes Research Institute (PCORI) outside the submitted work. 

DAL has received investigator-initiated educational grants from Bristol-Myers Squibb (BMS), has been a speaker for Bayer, Boehringer Ingeheim, and BMS/Pfizer and has consulted for BMS, and Boehringer Ingelheim.

HSL declares no funding conflict and honoraria from Abbott.

DQ declares no funding conflict and honoraria from Abbott.

MoH, SG, MSH, PD LM declare no funding conflicts

All authors report no conflicts of interest.

Figure 1 has been recoloured as suggested by Reviewer 2, and exported from GIMP as a tif, and it has been checked by PACE as suggested on the website.

Consent procedures: More information has been added to the methods section on the consent procedures (see page 26).

Table 1 has been reformatted although this was not requested, we feel the current format is easier to read (see page 17).

Data availability statement: Data is now provided as a supplementary file in an excel format. We have included the statement id but not the actual text as discussed before this is being used to develop a patient reported outcome measure as part of a larger substantive piece of work which is ongoing. It is hoped that we will be published next year (2023). We have included the similarity matrix data and distance matrix data which underpins the hierarchical cluster analysis and multi-dimensional scaling as well as the rating scale scores for frequency, importance and relevance.

There are no changes or retractions for the reference list. 

We hope the revisions to the manuscript meet with your approval and we look forward to hearing from you.

Your sincerely

Dr Anita Slade

Email: a.l.slade@bham.ac.uk

Response to Reviewers’ comments

Reviewer 1.

We would like to thank the Reviewer for their positive comments.

Reviewer #1: The authors present the results of a quite sophisticated study of a small group (18) of LVAD recipients who are being followed at one of the Heart Failure Canters in UK. The study targets eliciting and evaluating personal experience of living with an LVAD using Concept Group Mapping (GCM) approach. The participants were first asked to generate statements reflecting such experience in several categories. Such statements were then grouped, and the participants have rated those in three categories of relevance, frequency, and importance. All responses were then put through quite complex statistical analysis which evaluated several clusters and allowed comparisons between the clusters and inside each cluster using scores generated for such comparisons. The authors claim that this approach offers important advantage to other Patient Related Outcome Measures (PROMs) often used in evaluating outcomes of LVAD implantation or other life-modifying procedures. The main such advantage comes from the fact that the data are generated by allowing the actual recipients of the treatment (such as LVAD) to make statements describing different aspects of their life with the LVAD in their own words rather than answer prepared questions. There is data in the literature that the most accurate information regarding life with the LVAD comes from those who live with the LVAD. In addition, GCM approach has been successfully applied for concept generation and evaluation in the past. It does seem to have allowed the authors to benefit from working with the data generated by a small number of participants. Overall, the findings of the study do contribute some additional information as to is important to the LVAD recipients at which stage of their with the LVAD. However, it is hard to imagine that this information by itself would influence the direction of development of the new generations of LVADs.

Response to reviewer 1.

Thank you for your positive review of our paper. We agree that working with this group of LVAD recipients has given us an overview of their experiences of living with an LVAD, and their positive and negative experiences across a range of issues related to living with the LVAD. We agree that this part of the project itself will not influence the development of a new generation of LVADs but we hope some of the issues highlighted in the equipment and clothing section in particular may help us identify potential solutions with further research in the future. This project is also part of a larger project which has included in-depth qualitative interviews using GCM as a guide for semi-structured interviews. This has allowed us to explore the issues related to living with the LVAD in more depth. We hope that future development of patient reported outcome measures that are specific to LVAD recipients based on the GCM and interviews may address some of their concerns as well as supporting research and development of design evolutions. 

Reviewer 2

Reviewer #2: The implantation of a Left Ventricular Assist Device (LVAD) changes the life of both the patient and their family. Despite clinical improvement in most cases, the user is faced with equipment that imposes restrictions and changes in their image as a result of the surgery and the driveline. Beyond the usual knowledge about LVAD functioning and device-associated complications (such as infections, bleeding, thrombosis, or hemolysis), the attending physician should try to address the patient's feelings about their device. This manuscript does an excellent job of putting the user experience with LVADs at the center of the research, making it possible to identify domains and items that can be used in the development of Patient-Reported Outcome Measures (PROMs) in future research. Using the online Group Concept Mapping (GCM) software to identify these domains and items was also a great success for the authors, as it allowed people using LVADs who lived in different geographic regions to collaborate with the research. In addition to allowing the creation of statements, the GCM allows the research participant to sort and rate them, making it possible to identify what should be prioritized in future PROMS.

Response to Reviewer 2

We would like to thank the Reviewer for their positive comments.

Despite the good work done by the investigation team, we were able to list some recommendations to improve the understanding of the information generated by the research.

Figures and Tables:

· Figure 1A: Although the figure looks solid, the language of its legend is not clear, making it difficult to understand. I advise you to put some details on the interpretation of the legend as found in Line 209: "Statements frequently grouped together are located proximally, while less frequently grouped 210 statements are distal to each other."

Response to Reviewer 2

We have revisited all the figures and recoloured them to help with the differentiation between the different clusters, and added legends into the text as suggested. We have also included additional explanations in Figure 2 which we hope will make it easier to interpret. Unfortunately, because of the way the bridging and anchor statements figures are generated (within the GCM software) it has not been possible to recolour them. We tried to make them clearer using the online suggestions from PLOS one.

Fig 1. Legend

Legend Fig 1a Points maps are visual representations of the two-dimensional solution generated from the similarity matrix created from participants choices in stage 2. Frequently grouped statements are located proximally while less frequently grouped statements are distal to each other. (see page 12)

Figure 1B: This figure represents the clusters well; however, we suggest that each cluster had a different color to facilitate visualization. It’s interesting to add the description about the size of the clusters found between lines 258 - 260.

Legend Fig 1b Cluster maps are visual representations of the hierarchical cluster analysis solution and represent similarities and inter-relationships between the domains and items. Large clusters represent broader concepts, whereas smaller clusters represent more narrowly focused homogenous domains. (see page 12)

Figure 2: This figure is not clear. We suggest detailing a little more about bridging values in the legend (similar between lines 319 and 325). Also, the lines have similar colors, but different thicknesses, so you could be specified what each line thickness is and there are different color points without explanation.

We have included additional explanations in Figure 2 which we hope will make it easier to interpret. Unfortunately, because of the way the bridging and anchor statements figures are generated (within the GCM software) it has not been possible to recolour them. We tried to make them clearer using the online suggestions from PLOS one.

Figure 3: This figure is the most important, because it shows the importance, frequency, and relevance of each cluster. My suggestion is to put each cluster and line with different colors.

Response to Reviewer 2

We have revisited all the figures and recoloured them to help with the differentiation between the different clusters, and added legends into the text as suggested. 

Figure 2: This figure is not clear. We suggest detailing a little more about bridging values in the legend (similar between lines 319 and 325). Also, the lines have similar colors, but different thicknesses, so you could be specified what each line thickness is and there are different color points without explanation.

Response to Reviewer 2

Thank you for your comments we have added some explanation of the different thicknesses and what the different colour points mean to the figure. Additional information has been added to the legend in the manuscript.

Fig 2 legend

Bridging values ranging from 0 to 1 (Fig 2. Table 1) and indicate the extent to which sorting responses by participants were similar in stage 2. Higher values suggest a statement is more heterogeneous and has links to broader concepts (Fig 2a). While lower values closer to 0 suggest statements were more homogeneous and more frequently sorted together. These statements can be “anchors” for other statements close to them within the cluster (Fig 2b).[28] Spanning Values (Fig 2) represent how frequently statements were sorted together, a thicker line demonstrates that other statements were frequently sorted with the selected statement (100 ‘I have to look after my equipment as my life relies on it’ and 27 ‘Without my LVAD I wouldn’t be here’).

Figure 3: This figure is the most important, because it shows the importance, frequency, and relevance of each cluster. My suggestion is to put each cluster and line with different colors.

Response to Reviewer 2

Thank you for your comments, this has been recoloured to match the cluster map.

---

## [Decision Letter · Decision Letter 1]

3 Aug 2022

Living with a Left Ventricular Assist Device: capturing recipients experiences using group concept mapping softwareo:p/o:p

PONE-D-21-38323R1

Dear Dr. Slade,

We’re pleased to inform you that your manuscript has been judged scientifically suitable for publication and will be formally accepted for publication once it meets all outstanding technical requirements.

Kind regards,

Salil Deo

Academic Editor

PLOS ONE

Additional Editor Comments (optional):

Thank you very much for submitting your work to PLOS ONE. We are delighted to inform you that your paper has been accepted for publication.

Reviewers' comments:

Reviewer's Responses to Questions

**Comments to the Author**

1. If the authors have adequately addressed your comments raised in a previous round of review and you feel that this manuscript is now acceptable for publication, you may indicate that here to bypass the “Comments to the Author” section, enter your conflict of interest statement in the “Confidential to Editor” section, and submit your "Accept" recommendation.

Reviewer #2: All comments have been addressed

2. Is the manuscript technically sound, and do the data support the conclusions?

Reviewer #2: Yes

3. Has the statistical analysis been performed appropriately and rigorously? 

Reviewer #2: Yes

4. Have the authors made all data underlying the findings in their manuscript fully available?

Reviewer #2: Yes

5. Is the manuscript presented in an intelligible fashion and written in standard English?

Reviewer #2: Yes

6. Review Comments to the Author

Reviewer #2: The authors adequately responded to the reviewers' questions. I suggest that the article be accepted for publication.

7. PLOS authors have the option to publish the peer review history of their article (what does this mean?). If published, this will include your full peer review and any attached files.

Reviewer #2: No

---

## [Editor Report · Acceptance letter]

6 Sep 2022

PONE-D-21-38323R1 

Living with a Left Ventricular Assist Device: capturing recipients experiences using group concept mapping software 

Dear Dr. Slade:

I'm pleased to inform you that your manuscript has been deemed suitable for publication in PLOS ONE. Congratulations! Your manuscript is now with our production department. 

Kind regards, 

on behalf of

Dr. Salil Deo 

Academic Editor

PLOS ONE